# Policy Gradient Algorithms
# Implicitly Optimize by Continuation

**Adrien Bolland**  *adrien.bolland@uliege.be*
*Montefiore Institute, University of Liège*

**Gilles Louppe**  *g.louppe@uliege.be*
*Montefiore Institute, University of Liège*

**Damien Ernst**  *dernst@uliege.be*
*Montefiore Institute, University of Liège*
*LTCI, Telecom Paris, Institut Polytechnique de Paris*

**Reviewed on OpenReview:** *https://openreview.net/forum?id=3Ba6Hd3nZt*

## Abstract

Direct policy optimization in reinforcement learning is usually solved with policy-gradient algorithms, which optimize policy parameters via stochastic gradient ascent. This paper provides a new theoretical interpretation and justification of these algorithms. First, we formulate direct policy optimization in the optimization by continuation framework. The latter is a framework for optimizing nonconvex functions where a sequence of surrogate objective functions, called continuations, are locally optimized. Second, we show that optimizing affine Gaussian policies and performing entropy regularization can be interpreted as implicitly optimizing deterministic policies by continuation. Based on these theoretical results, we argue that exploration in policy-gradient algorithms consists in computing a continuation of the return of the policy at hand, and that the variance of policies should be history-dependent functions adapted to avoid local extrema rather than to maximize the return of the policy.

## 1 Introduction

Applications where one has to control an environment are numerous and solving these control problems efficiently is the preoccupation of many researchers and engineers. Reinforcement learning (RL) has emerged as a solution when the environments at hand have complex and stochastic dynamics (Sutton & Barto, 2018). Direct policy optimization and more particularly (on-policy) policy gradients are methods that have been successful in recent years. These methods, reviewed by Duan et al. (2016) and Andrychowicz et al. (2020), all consist in parameterizing a policy (most often with a neural network) and adapting the parameters with a local-search algorithm in order to maximize the expected sum of rewards received when the policy is executed, called the return of the policy. We distinguish two basic elements that determine the performance of these methods. As first element, we have the formalization of the optimization problem. It is defined through two main choices: the (functional) parametrization of the policy and the learning objective function, which mostly relies on adding an entropy regularization term to the return. As second element, there is the choice of the local-search algorithm to solve the optimization problem – we focus on stochastic gradient ascent methods in this study.

The policy parameterization is the first formalization choice. In theory, there exists an optimal deterministic policy (Sutton & Barto, 2018), which can be optimized by deterministic policy gradient (Silver et al., 2014) with a guarantee of converging towards a stationary solution (Xiong et al., 2022). However, this approach may give poor results in practice as it is subject to convergence towards local optima (Silver et al., 2014).

It is therefore usual to optimize stochastic policies where this problem is mitigated in practice (Duan et al., 2016; Andrychowicz et al., 2020). For discrete state and action spaces, theoretical guarantees of global convergence hold for softmax or direct policy parameterization (Bhandari & Russo, 2019; Zhang et al., 2021; Agarwal et al., 2020). In the general case of continuous spaces, these results no longer hold and only convergence towards stationarity can be ensured under strong hypotheses (Bhatt et al., 2019; Zhang et al., 2020b; Bedi et al., 2021). Recently, convergence under milder assumptions was established assuming that the policy follows a heavy-tailed distribution, which guarantees a sufficiently spread distribution of actions (Bedi et al., 2022). Nevertheless, most of the empirical works have focused on (light-tailed) Gaussian policies (Duan et al., 2016; Andrychowicz et al., 2020) for which convergence is thus not ensured in the general case (Bedi et al., 2022). The importance of a sufficiently spread distribution in policy gradient had already been observed in early works and was loosely interpreted as exploration (Lillicrap et al., 2015; Mnih et al., 2016). This concept originally introduced in bandit theory and value-based RL, where it consists in selecting a suboptimal action to execute in order to refine a statistical estimate (Simon, 1955; Sutton & Barto, 2018), is to our knowledge not well defined for direct policy optimization. Other empirical works also showed that relying on Beta distributions when the set of actions is bounded within an interval outperformed Gaussian policies (Chou et al., 2017; Fujita & Maeda, 2018). As a side note, another notable advantage of stochastic policies is the possibility to rely on information geometry and use efficient trust-region methods to speed up the local-search algorithms (Shani et al., 2020; Cen et al., 2022). In summary, no consensus has yet been reached on the exact policy parameterization that should be used in practice.

The second formalization choice is the learning objective and more particularly the choice of entropy regularization. Typically, a bonus enforcing the uniformity of the action distribution is added to the rewards in the objective function (Williams & Peng, 1991; Haarnoja et al., 2019). Intuitively, it avoids converging too fast towards policies with small spread, which are subject to being locally optimal. More general entropy regularizations were applied for encouraging high-variance policies while keeping the distribution sparse (Nachum et al., 2016) or enforcing the uniformity of the state-visitation distribution in addition to the action distribution (Islam et al., 2019). Again, no consensus is reached about the best regularization to use in practice.

The importance of introducing sufficient stochasticity and regularizing entropy is commonly accepted in the community. Some preliminary research has been conducted to develop a theoretical foundation for this observation. Ahmed et al. (2019) proposed an empirical analysis of the impact of the entropy regularization term. They concluded that adding this term yields a smoothed objective function. A local-search algorithm will therefore be less prone to convergence to local optima. This problem was also studied by Husain et al. (2021). They proved that optimizing a policy by regularizing the entropy is equivalent to performing a robust optimization against changes in the reward function. This result was recently reinterpreted by Brekelmans et al. (2022) who deduced that the optimization is equivalent to a game where one player adapts the policy while an adversary adapts the reward. The research papers that have been reviewed concentrate solely on learning objectives in the context of entropy regularization, leaving unanswered the question of the relationship between a policy's return and the distribution of actions. This question is of paramount importance for understanding how the formalization of the direct policy optimization problem impacts the resulting control strategy.

In this work, we propose a new theoretical interpretation of the effects of the action distribution on the objective function. Our analysis is based on the theory of optimization by continuation (Allgower & Georg, 1980), which consists in locally optimizing a sequence of surrogate objective functions. The latter are called continuations and are often constructed by filtering the optimization variables in order to remove local optima. Our main contributions are twofold. First, we define a continuation for the return of policies and formulate direct policy optimization in the optimization by continuation framework. Second, based on this framework, we study different formulations, i.e., policy parameterization and entropy regularization, of direct policy optimization. Several conclusions are drawn from the analysis. First, we show that the continuation of the return of a deterministic policy is equal to the return of a Gaussian policy. Second, we show that the continuation of the return of a Gaussian policy equals the return of another Gaussian policy with scaled variance. We then derive from the previous results that optimizing Gaussian policies using policy-gradient algorithms and performing regularization can be interpreted as optimizing deterministic policies by

continuation. In this regard, exploration as it is usually understood in policy gradients, consists in computing the continuation of the return of the policy at hand. Finally, we show that for a more general continuation, the continuation of the return of a deterministic policy equals the return of a Gaussian policy where the variance is a function of the observed history of states and actions. These results provide a new interpretation for the variance of a policy: it can be seen as a parameter of the policy-gradient algorithm instead of an element of the policy parameterization. Moreover, to fully exploit the power of continuations, the variance of a policy should be a history-dependent function iteratively adapted to avoid the local extrema of the return.

Optimization by continuation provides or aims to provide two main practical advantages to solve optimization problems. First, these methods smooth the objective function and allow the application of gradient-based optimization methods, as discussed in the literature on variational optimization (Staines & Barber, 2012) and applied by Murray & Ng (2010) to discrete optimization problems. Second, it enables to compute the global optimum of the optimization problem. This global optimum can be reached for example assuming that the optimum of the first continuation is simple to compute, and that the path of local optima of each continuation leads to the global optimum of the problem (Allgower & Georg, 1980). Theoretical guarantees on convergence are still scarce in the literature. Several works focus on Gaussian continuations where the continuations are convolutions of the objective function by Gaussian kernels (Mobahi & Fisher III, 2015). It is particularly useful when the objective function is concave, ensuring smoothness of the surrogate optimization problems and providing convergence guarantees to the global optimal solution (Nesterov & Spokoiny, 2017). Interestingly, a recent work links these continuations to concave envelopes (Mobahi & Fisher, 2015). Another notable result holds for similar continuations relying on uniform kernels for building the surrogate problems where convergence to the global solution is guaranteed under certain assumptions on the objective function (Hazan et al., 2016). In the general case, finding the right sequence of continuations for achieving convergence to the global optimum remains heuristic and problem-dependent. The framework of optimization by continuation can nevertheless provide valuable insights, as will be further explored in this work.

Despite the lack of theoretical guarantees, optimization by continuation has found successful machine learning applications, including image alignment (Mobahi et al., 2012), greedy layerwise training of neural networks (Bengio, 2009), and neural network training by iteratively increasing the non-linearity of the activation functions (Pathak & Paffenroth, 2019). To our knowledge, optimization by continuation has never yet been applied to direct policy optimization. However, optimizing a distribution over the policy parameters rather than directly optimizing the policy is a reinforcement learning technique that has been used to perform direct policy optimization (Sehnke et al., 2010; Salimans et al., 2017; Zhang et al., 2020a). Among other things, it decreases the variance of the gradient estimates in some cases. If this distribution over policy parameters is a Gaussian, it is furthermore by definition equivalent to optimizing the policy by Gaussian continuation (Mobahi & Fisher III, 2015). Another method, called reinforcement learning with logistic reward-weighted regression (Wierstra et al., 2008; Peters & Schaal, 2007), consists in optimizing a surrogate objective of the return. The surrogate is the expected utility of the sum of rewards. Originally justified relying on the field of decision theory (Chernoff & Moses, 2012), it can equivalently be seen as an optimization by continuation method.

The paper is organized as follows. In Section 2, the background of direct policy optimization is reminded. The framework for optimizing policies by continuation is developed in Section 3 and theoretical results relating the return of policies to their continuations are presented in Section 4. In Section 5, these results are used for elaborating on the formulations of direct policy optimization. Finally, the results are summarized and further works discussed in Section 6.

## 2 Theoretical Background

In this section, we remind the background of reinforcement learning in Markov decision processes and discuss the direct policy optimization problem.

## 2.1 Markov Decision Processes

We study problems in which an agent makes sequential decisions in a stochastic environment in order to maximize an expected sum of rewards (Sutton & Barto, 2018). The environment is modeled with an infinite-time Markov Decision Process (MDP) composed of a state space $\mathcal{S}$, an action space $\mathcal{A}$, an initial state distribution with density $p_0$, a transition distribution (dynamic) with conditional density $p$, a bounded reward function $\rho$, and a discount factor $\gamma \in [0, 1[$. When an agent interacts with the MDP $(\mathcal{S}, \mathcal{A}, p_0, p, \rho, \gamma)$, first, an initial state $s_0 \sim p_0(\cdot)$ is sampled, then, the agent provides at each time step $t$ an action $a_t \in \mathcal{A}$ leading to a new state $s_{t+1} \sim p(\cdot|s_t, a_t)$. Such a sequence of states and actions $h_t = (s_0, a_0, \ldots, s_{t-1}, a_{t-1}, s_t) \in H$ is called a history and $H$ is the set of all histories of any arbitrary length. In addition, at each time step $t$, a reward $r_t = \rho(s_t, a_t) \in \mathbb{R}$ is observed.

A (stochastic) history-dependent policy $\eta \in \mathcal{E} = H \to \mathcal{P}(\mathcal{A})$ is a mapping from the set of histories $H$ to the set of probability measures on the action space $\mathcal{P}(\mathcal{A})$, where $\eta(a|h)$ is the associated conditional probability density of action $a$ given the history $h$. A (stochastic) Markov policy $\pi \in \Pi = \mathcal{S} \to \mathcal{P}(\mathcal{A})$ is a mapping from the state space $\mathcal{S}$ to the set of probability measures on the action space $\mathcal{P}(\mathcal{A})$, where $\pi(a|s)$ is the associated conditional probability density of action $a$ in state $s$. Finally, deterministic policies $\mu \in M = \mathcal{S} \to \mathcal{A}$ are functions mapping an action $a = \mu(s) \in \mathcal{A}$ to each state $s \in \mathcal{S}$. We note that for each deterministic policy $\mu$ there exists an equivalent Markov policy, where the probability measure is a Dirac measure on the action $a = \mu(s)$ in each state $s$. In addition, for each Markov policy, there exists an equivalent history-dependent policy only accounting for the last state in the history. We therefore write by abuse of notation that $M \subsetneq \Pi \subsetneq \mathcal{E}$.

The function $J : \mathcal{E} \to \mathbb{R}$ is defined as the function mapping to any policy $\eta$ the expected discounted cumulative sum of rewards gathered by an agent interacting in the MDP by sampling actions from the policy $\eta$. The value $J(\eta)$ is called the return of the policy $\eta$ and is computed as follows:

$$J(\eta) = \mathbb{E}_{\substack{s_0 \sim p_0(\cdot) \\ a_t \sim \eta(\cdot|h_t) \\ s_{t+1} \sim p(\cdot|s_t, a_t)}} \left[ \sum_{t=0}^{\infty} \gamma^t \rho(s_t, a_t) \right] . \tag{1}$$

An optimal agent follows an optimal policy $\eta^*$ maximizing the expected discounted sum of rewards $J$.

## 2.2 Direct Policy Optimization

**Problem statement.** Let $(\mathcal{S}, \mathcal{A}, p_0, p, \rho, \gamma)$ be an MDP and let $\eta_\theta \in \mathcal{E}$ be a policy parameterized by the real vector $\theta \in \mathbb{R}^{d_\Theta}$. The objective of the optimization problem is to find the optimal parameter $\theta^* \in \mathbb{R}^{d_\Theta}$ such that the return of the policy is maximized:

$$\theta^* = \underset{\theta \in \mathbb{R}^{d_\Theta}}{\operatorname{argmax}} \, J(\eta_\theta) . \tag{2}$$

In this work, we consider on-policy policy-gradient algorithms (Andrychowicz et al., 2020). These algorithms optimize differentiable policies with local-search methods using the derivatives of the policies. They iteratively repeat two operations. First, they approximate an ascent direction relying on histories sampled from the policy, with the current parameters, in the MDP. Second, they update these parameters in the ascent direction.

**Deterministic Policies.** It is possible to approximate a solution of the optimization problem equation (2) for a deterministic policy parameterized with a function approximator $\mu_\theta \in M$ by stochastic gradient ascent using the deterministic policy gradient theorem (Silver et al., 2014). Nevertheless, optimizing deterministic policies with *inadequate exploration* (i.e., without sufficient additional policy randomization) usually results in locally optimal policies with poor performance (Silver et al., 2014).

**Gaussian Policies.** In direct policy optimization, most of the works focus on learning Markov policies where the actions follow a Gaussian distribution whose mean and covariance matrix are parameterized

function approximators (Duan et al., 2016; Andrychowicz et al., 2020). More precisely, a parametrized Gaussian policy $\pi_\theta^{GP} \in \Pi$ is a policy where the actions follow a Gaussian distribution of mean $\mu_\theta(s)$ and covariance matrix $\Sigma_\theta(s)$ for each state $s$ and parameter $\theta$, it thus has the following density:

$$\pi_\theta^{GP}(a|s) = \mathcal{N}(a|\mu_\theta(s), \Sigma_\theta(s)) . \tag{3}$$

**Affine Policies.** A parameterized policy (deterministic or stochastic) is said to be affine, if the function approximators used to construct the functional form of the policy are affine functions of the parameter $\theta$. Formally, each function approximator $f_\theta$ of a history-dependent policy has the following form $\forall h \in H$:

$$f_\theta(h) = a(h)^T \theta + b(h) , \tag{4}$$

where $a$ and $b$ are functions building features from the histories. Such policies are often considered in theoretical studies (Busoniu et al., 2017) and perform well on complex tasks in practice (Rajeswaran et al., 2017).

## 3 Optimizing Policies by Continuation

In this section, we introduce the optimization by continuation methods and formulate direct policy optimization in this framework.

### 3.1 Optimization by Continuation

Optimization by continuation (Allgower & Georg, 1980) is a technique used to optimize nonconvex functions with the objective of avoiding local extrema. A sequence of optimization problems is solved iteratively using the optimum of the previous iteration. Each problem consists in optimizing a deformation of the original function and is typically solved by local search. Through the iterations, the function is less and less deformed. Such procedure is also sometimes referred to as graduated optimization (Blake & Zisserman, 1987) or optimization by homotopy (Watson & Haftka, 1989), as the homotopy of a function refers to its deformation in topology.

Formally, let $f : \mathcal{X} \to \mathbb{R}$ be the real-valued function to optimize. Let $g : \mathcal{Y} \to \mathbb{R}$ be another real-valued function used for building the deformation of $f$. Finally, let the conditional distribution function $p : \mathcal{X} \to \mathcal{P}(\mathcal{Y})$ be the mapping from an optimization variable $x \in \mathcal{X}$ to the set of probability measures $\mathcal{P}(\mathcal{Y})$, such that $p(y|x)$ is the associated density function for any random event $y \in \mathcal{Y}$ given $x \in \mathcal{X}$. The continuation of the function $f$ under the distribution $p$ and deformation function $g$ is defined as the function $f^p : \mathcal{X} \to \mathbb{R}$ such that $\forall x \in \mathcal{X}$:

$$f^p(x) = \mathbb{E}_{y \sim p(\cdot|x)} [g(y)] . \tag{5}$$

For the optimization by continuation described hereafter, there must exist a conditional distribution $p^*$ for which $f^p$ equals $f$ in the limit as $p$ approaches $p^*$. A typical example is to choose the function $g$ equal to $f$, and to use a Gaussian distribution with a constant diagonal covariance matrix for the distribution $p$. We then have so-called Gaussian continuations (Mobahi & Fisher III, 2015).

Finally, optimizing a function $f$ by continuation involves iteratively locally optimizing its continuation for a sequence of conditional distributions approaching $p^*$ with decreasing spread. Formally, let $p_0 \succ p_1 \succ \cdots \succ p_{I-1}$ be a sequence of conditional distributions (monotonically) approaching $p^*$ with strictly decreasing covariance matrices[1]. Then, optimizing $f$ by continuation consists in locally optimizing its continuation $f^{p_i}$ with a local-search algorithm initialized at $x_i^*$ for each iteration $i$. This general procedure is summarized in Algorithm 1. Particular instances of this algorithm are described by Hazan et al. (2016) and Shao et al. (2019) for Gaussian continuations.

---

[1]In this work, we consider the $L^2$-norm of functions and the Loewner order over the set of covariance matrices (Siotani, 1967).

In practice, the optimization process can be approximated by performing a limited number of local-search iterations at each step of the optimization by continuation. In the following sections, we consider that each optimization of the continuation $f^{p_i}$ is approximated with a single gradient ascent step and that the continuation distribution sequence $p_0 \succ p_1 \succ \cdots \succ p_{I-1}$ is constructed by iteratively reducing the variance of the distribution $p_i$. Note that if this variance reduction is sufficiently slow, and the stepsize is well chosen, a single gradient ascent step enables to accurately approximate $x_i^*$.

---

**Algorithm 1** Optimization by Continuation

---

1: Provide a sequence of $I$ functions $p_0 \succ p_1 \succ \cdots \succ p_{I-1}$
2: Provide an initial variable value $x_0^* \in \mathcal{X}$ for the local search
3: **for** $i = 0, 1, \ldots, I - 1$ **do**
4: $\quad x_{i+1}^* \leftarrow$ Optimize the continuation $f^{p_i}$ by local search initialized at $x_i^*$
5: **end for**
6: **return** $x_I^*$

---

### 3.2 Continuation of the Return of a Policy

The direct policy optimization problem usually consists in maximizing a nonconvex function. Optimization by continuation is thus a good candidate for computing a solution. In this section, we introduce a novel continuation adapted to the return of policies.

The return of a policy depends on the probability of a sequence of actions through the product of the density $\eta_\theta(a_t|s_t)$ of each action $a_t$ for a given parameter $\theta$, see equation (1). We define the continuation of interest as the expectation of the return where each factor in the product of densities depends on a different parameter vector. This expectation is taken according to a distribution that disturbs these parameter vectors at each time step with a variance depending on the history. Formally, using the notations from Section 3.1, we optimize the function $f$ that for all $x = \theta$ equals the return, $f(\theta) = J(\pi_\theta)$, over the set $\mathcal{X} = \mathbb{R}^{d_\Theta}$. Let the covariance function $\Lambda : H \to \mathbb{R}^{d_\Theta \times d_\Theta}$ be a function mapping a history $h_t \in H$ to a covariance matrix $\Lambda(h_t)$. Let the continuation distribution $q$ be a distribution such that $q(\theta_t|\theta, \Lambda(h_t))$ is the density of $\theta_t$ distributed with mean $\theta$ and covariance matrix $\Lambda(h_t)$. Then, let $\mathcal{Y} = (\mathcal{S} \times \mathcal{A} \times \mathbb{R}^{d_\Theta})^{\mathbb{N}}$ be the set of (infinite) sequences of states, actions and parameters and let $p$ and $g$, the two functions defining the continuation, be as follows:

$$p(y|x) = p(s_0) \prod_{t=0}^{\infty} \eta_{\theta_t}(a_t|h_t) p_\theta(\theta_t|h_t) p(s_{t+1}|s_t, a_t) \tag{6}$$

$$g(y) = \sum_{t=0}^{\infty} \gamma^t \rho(s_t, a_t) , \tag{7}$$

where $p_\theta(\theta_t|h_t) = q(\theta_t|\theta, \Lambda(h_t))$ such that the spread of $p_\theta$ depends on the function $\Lambda$. Taken together, the continuation $f_\Lambda^q = f^p$ of the return of the policy $\eta_\theta \in \mathcal{E}$ corresponding to the distribution $q$ and covariance function $\Lambda$, is defined $\forall \theta \in \mathbb{R}^{d_\Theta}$ as:

$$f_\Lambda^q(\theta) = \mathbb{E}_{\substack{s_0 \sim p_0(\cdot) \\ \theta_t \sim q(\cdot|\theta, \Lambda(h_t)) \\ a_t \sim \eta_{\theta_t}(\cdot|h_t) \\ s_{t+1} \sim p(\cdot|s_t, a_t)}} \left[ \sum_{t=0}^{\infty} \gamma^t \rho(s_t, a_t) \right] . \tag{8}$$

Finally, the continuation equation (8) converges towards the return of $\eta_\theta$ in the limit as the covariance function $\Lambda$ approaches zero, as required in Section 3.1.

This continuation is expected to be well-suited for removing local extrema of the return for three main reasons. First, marginalizing the variables of a function as in our continuation is expected to smooth this function and therefore remove local extrema – the particular case of Gaussian blurring has been widely studied in the literature (Mobahi & Fisher, 2015; Nesterov & Spokoiny, 2017). Second, we underline the interest of considering a continuation in which the disturbance of the policy parameters may vary based on

the time step. Indeed, changing the parameter vector of the policy at different time steps (and changing the action distributions) may modify the objective function in significantly different ways. Third, we justify the factorization of the conditional distribution $p_\theta$ equation (6) by the causal effect of actions in the MDP. As the actions only influence the rewards to come, the past history is expected to provide a sufficient statistic for disturbing the parameters in order to remove local optima. We therefore chose parameter probabilities conditionally independent given the past history. This history-dependency is encoded through the covariance function $\Lambda$ in equation (8).

Maximizing $f_\Lambda^q$ to solve the optimization problem from Algorithm 1 is a complicated task. A common local-search algorithm used in machine learning is stochastic gradient ascent, which is known for performing well on several complex functions depending on many variables (Bottou, 2010). The gradient of $f_\Lambda^q$ can be computed by Monte-Carlo sampling applying the reparameterization trick (Goodfellow et al., 2016) for simple continuation distributions or relying on the REINFORCE trick (Williams, 1992) in the more general case. Due to the complex time dependencies of the random events, these vanilla gradient estimates have practical limitations: the estimates may have large variance, the infinite horizon shall be truncated, and the direction provided is computed in the Euclidean space of parameters rather than in a space of distributions (Peters & Schaal, 2008). Finally, the evaluation of the continuation and its derivatives require one to sample parameter vectors, which may be computationally expensive for complex high-dimensional distributions. The study of different continuation distributions and the application of the optimization procedure from Algorithm 1 to practical problems is left for further works. In this work, we rather rely on the continuation to study existing direct policy optimization algorithms. To this end, we show in the next section that maximizing the continuation defined by equation (8) is equivalent to solving a direct policy optimization problem for another policy, called a mirror policy.

## 4 Mirror Policies and Continuations

This section is dedicated to the interpretation of the continuation of the return of a policy. We show it equals the return of another policy, called a mirror policy. The existence and closed form of mirror policies is also discussed.

### 4.1 Optimizing by Continuation with Mirror Policies

**Definition 1.** *Let $(\mathcal{S}, \mathcal{A}, p_0, p, \rho, \gamma)$ be an MDP and let $\eta_\theta \in \mathcal{E}$ be a history-dependent policy parameterized with the vector $\theta \in \mathbb{R}^{d_\Theta}$. In addition, let $f_\Lambda^q$ be the continuation of the return of the policy $\eta_\theta$ corresponding to a continuation distribution $q$ and covariance function $\Lambda$ as defined in equation (8). We call a mirror policy of the original policy $\eta_\theta$, under the continuation distribution $q$ and covariance function $\Lambda$, any history-dependent policy $\eta_\theta' \in \mathcal{E}$ such that $\forall \theta \in \mathbb{R}^{d_\Theta}$:*

$$f_\Lambda^q(\theta) = J(\eta_\theta') \,. \tag{9}$$

Let us assume we are provided with the continuation $f_\Lambda^q$ of the return of an original policy $\eta_\theta$ depending on the parameter $\theta$ that shall be optimized. In addition, let us assume we can compute a mirror policy $\eta_\theta'$ for the original policy $\eta_\theta$. By Definition 1, the continuation of the original policy equals the return of the mirror policy for all $\theta$. In addition, under smoothness assumptions, all their derivatives are equal too. Therefore, maximizing the continuation of an original policy by stochastic gradient ascent can be performed by maximizing the return of its mirror policy by policy gradient. Applying state-of-the-art policy-gradient algorithms on the mirror policies for optimizing the continuations in Algorithm 1 may alleviate several of the shortcomings of the optimization procedure described earlier.

### 4.2 Existence and Closed Form of Mirror Policies

In this section, we first show that there always exists a mirror policy. In addition, several closed forms are provided depending on the original policy, the continuation distribution, and the covariance function.

**Theorem 1.** *For any original history-dependent policy $\eta_\theta \in \mathcal{E}$ parameterized with the vector $\theta \in \mathbb{R}^{d_\Theta}$ and for any continuation distribution $q$ and covariance function $\Lambda$, there exists a mirror history-dependent policy*

$\eta'_\theta \in \mathcal{E}$ of the original policy $\eta_\theta$ that writes as:

$$\eta'_\theta(a|h) = \underset{\theta' \sim q(\cdot|\theta, \Lambda(h))}{\mathbb{E}} [\eta_{\theta'}(a|h)] \ . \tag{10}$$

Theorem 1 guarantees the existence of mirror policies. Such a mirror policy is a function depending on the same parameters as its original policy but that has a different functional form and may therefore provide actions following a different distribution compared to the original policy.

Theorem 1 leads to two important corollary results. First, as demonstrated in Theorem 2 in Appendix A, let $\eta''$ be a mirror policy of $\eta'$ and let $\eta'$ be a mirror policy of the original policy $\eta$ of the form of equation (10). Then, there exists a continuation for which $\eta''$ is a mirror policy of the original policy $\eta$. It follows that the return of the mirror policy of another mirror policy is itself equal to a continuation of the original policy. Second, Theorem 1 also reveals that for a given original policy and continuation distribution, the variance of the mirror policy is defined through the continuation covariance function $\Lambda$. Furthermore, we remind that the variance of the continuation is an hyperparameter that shall be selected for each iteration of the optimization by continuation, see Section 3. This choice of hyperparameter is thus reflected as the choice of the variance of a mirror policy. The expert making this choice sees the effect of the disturbed parameters on the environment through the variance of the mirror policy. From a practical perspective, it is probably easier to quantify the effect on the local extrema depending on the variance of the mirror policy rather than depending on the variance of the continuation.

**Property 4.1.** *Let the original policy $\pi_\theta \in \Pi$ be a Markov policy and let the covariance function depend solely on the last state in the history. Then, there exists a mirror Markov policy $\pi'_\theta \in \Pi$.*

Property 4.1 is an intermediate result providing sufficient assumptions on the continuation for having mirror Markov policies. Note that for this type of continuation, the parameters of the policy are disturbed independently of the history followed by the agent.

**Property 4.2.** *Let the original policy $\pi_\theta^{GP} \in \Pi$ be a Gaussian policy as defined in equation (3) with affine function approximators. Let the covariance function depend solely on the last state in the history and let the distribution $q$ be a Gaussian distribution. Then, there exists a mirror Markov policy $\pi'_\theta \in \Pi$ such that for all states $s \in \mathcal{S}$, it converges towards a Gaussian policy in the limit as the affine coefficients of the covariance matrix $\Sigma_\theta(s)$ approaches zero ($\|\nabla_\theta \Sigma_\theta(s)\| \to 0$):*

$$\pi'_\theta(a|s) \to \mathcal{N}(a|\mu_\theta(s), \Sigma'_\theta(s)) \ , \tag{11}$$

*where $\Sigma'_\theta(s) = C_\theta(s) + \Sigma_\theta(s)$ and $C_\theta(s) = \nabla_\theta \mu_\theta(s)^T \, \Lambda(s) \, \nabla_\theta \mu_\theta(s)$.*

Under the assumptions of Property 4.2, a mirror policy can be approached by a policy that only differs from the original one by having a variance which is increased by the term $C_\theta(s)$ proportional to the variance of the continuation. In particular, when the variance of the original policy $\pi_\theta^{GP}$ is solely dependent on the state, then $\|\nabla_\theta \Sigma_\theta(s)\| = 0$ and $\pi'_\theta(a|s) = \mathcal{N}(a|\mu_\theta(s), \Sigma'_\theta(s))$. In this case, for any $\theta$, the covariance matrix of this mirror policy is additionally bounded from below such that $\Sigma'_\theta(s) \succeq C_\theta(s)$.

**Property 4.3.** *Let the original policy $\mu_\theta \in M$ be an affine deterministic policy. Let the covariance function depend solely on the last state in the history and let the distribution $q$ be a Gaussian distribution. Then, the Markov policy $\pi_\theta^{GP'} \in \Pi$ is a mirror policy:*

$$\pi_\theta^{GP'}(a|s) = \mathcal{N}(a|\mu_\theta(s), \Sigma'_\theta(s)) \ , \tag{12}$$

*where $\Sigma'_\theta(s) = \nabla_\theta \mu_\theta(s)^T \, \Lambda(s) \, \nabla_\theta \mu_\theta(s)$.*

Therefore, under some assumptions, disturbing a deterministic policy and optimizing it afterwards can be interpreted as optimizing the continuation of the return of this policy.

**Property 4.4.** *Let the original policy $\mu_\theta \in M$ be an affine deterministic policy. Let the distribution $q$ be a Gaussian distribution. Then, the policy $\eta'_\theta \in \mathcal{E}$ is a mirror policy:*

$$\eta'_\theta(a|h) = \mathcal{N}(a|\mu_\theta(s), \Sigma'_\theta(h)) \ , \tag{13}$$

*where $\Sigma'_\theta(h) = \nabla_\theta \mu_\theta(s)^T \Lambda(h) \nabla_\theta \mu_\theta(s)$.*

Property 4.4 extends Property 4.3 to more general continuation distributions. This extension is used later to justify the interest of optimizing history-dependent policies in order to optimize an underlying deterministic policy by continuation. The theorem and properties are shown in Appendix A.

# 5 Implicit Optimization by Continuation

In this section, two formulations, i.e., a parameterized policy and a learning objective each, used by several policy-gradient algorithms are analyzed relying on original and mirror policies. In Section 5.1, we show that optimizing each formulation by local search corresponds to optimizing a continuation. The optimized policy is thus the mirror policy of an unknown original policy. We show the existence of the corresponding continuation and original policy and discuss their closed form. This analysis provides a novel interpretation of the state-of-the-art algorithms for direct policy optimization. We discuss the role of stochastic policies in light of this interpretation in Section 5.2.

## 5.1 Gaussian Policies and Regularization

The policy-gradient literature has mainly focused on optimizing two problem formulations by local search – typically with stochastic gradient ascent and (approximate) trust-region methods. First, the vast majority of works focuses on optimizing the return of Gaussian policies (Duan et al., 2016; Andrychowicz et al., 2020). Second, in many formulations this objective function is extended by adding a bonus to the entropy of the optimized policy (Williams & Peng, 1991; Haarnoja et al., 2019). We show that when optimizing a policy according to these formulations, there exists an (unknown) deterministic original policy and a continuation under which the optimized policy is a mirror policy. Provided with the local-search algorithm from the policy-gradient method, we conclude that optimizing both formulations is equivalent to implicitly optimizing a deterministic policy by continuation.

First, we remind that under Property 4.3, for any affine deterministic policy $\mu_\theta$, there exists an affine Gaussian mirror policy $\pi_\theta^{GP'}$ as defined by equation (12). In Property 5.1, the converse of Property 4.3 is stated, which answers to the question: *under which conditions a Gaussian policy is the mirror policy of an (unknown) deterministic policy.* For this converse statement to be true, the transformation between covariance functions in Property 4.3 must be surjective, which is guaranteed if $d_\mathcal{A} \leq d_\Theta$ and $\nabla_\theta \mu_\theta(s)$ is full rank. The first assumption is always met in practice and the second is met when no action is a deterministic function of the others.

**Property 5.1.** *Let $\pi_\theta^{GP'}$ be an affine Gaussian policy with mean function $\mu_\theta$, and with covariance function $\Sigma'_\theta = \Sigma'$ constant with respect to the parameters of the policy (i.e., a function depending solely on the state). If $d_\mathcal{A} \leq d_\Theta$ and if $\nabla_\theta \mu_\theta(s)$ is full rank, then, there exists a continuation, with covariance $\Lambda$ proportional to $\Sigma'$, for which $\pi_\theta^{GP'}$ is a mirror policy of the original policy $\mu_\theta$.*

Entropy regularization ensures that the variance of the policy remains sufficiently large during the optimization process.[2] Similar objectives are pursued with maximum entropy reinforcement learning (Haarnoja et al., 2019) or with (approximate) trust-region methods where the trust-region constraint is dualized (Schulman et al., 2015; 2017). Let us consider an affine Gaussian original policy $\pi_\theta^{GP}$ with constant covariance $\Sigma_\theta = \Sigma$. Under Property 4.2, there exists another affine Gaussian policy $\pi_\theta^{GP'}$ that is a mirror policy of $\pi_\theta^{GP}$. This mirror policy has the same mean function and a covariance function bounded from below by $C_\theta = C$. Property 5.2 provides the converse and answers to the question: *under which conditions a Gaussian policy with sufficiently large covariance is the mirror policy of an (unknown and Gaussian) policy.* Similar to the previous property, this is guaranteed when $d_\mathcal{A} \leq d_\Theta$ and $\nabla_\theta \mu_\theta(s)$ is full rank.

---

[2]Formally, for two matrices $A$ and $B$, we have that $A \succeq B \Rightarrow |A| \geq |B|$ (Siotani, 1967). As the entropy of a Gaussian policy is a concave function of the determinant of the covariance matrix, a bounded covariance matrix implies a bounded entropy. The entropy-regularization learning objective can therefore be interpreted as the Lagrangian relaxation of the latter entropy-bounded optimization problem.

**Property 5.2.** *Let $\pi_\theta^{GP'}$ be an affine Gaussian policy with mean function $\mu_\theta$, and with covariance function $\Sigma'_\theta = \Sigma' \succeq C$ constant with respect to the parameters of the policy (i.e., a function depending solely on the state) and bounded from bellow by $C$. If $d_\mathcal{A} \leq d_\Theta$ and if $\nabla_\theta \mu_\theta(s)$ is full rank, then, there exists a continuation, with covariance $\Lambda$ proportional to $C$, for which $\pi_\theta^{GP'}$ is a mirror policy of an original Gaussian policy $\pi_\theta^{GP}$ with the same mean function $\mu_\theta$ and with constant covariance function $\Sigma \preceq \Sigma'$.*

The two previous properties indicate that a Gaussian policy is guaranteed to be a mirror policy of another policy, Gaussian or deterministic, under some assumptions. If we furthermore guarantee that the continuation covariance decreases during the optimization, policy-gradient algorithms optimizing affine Gaussian policies can be interpreted as algorithms optimizing an original policy by continuation.

Let us consider two cases, each corresponding to a problem formulation, where we optimize by policy gradient an affine Gaussian policy $\pi_\theta^{GP'}$ with covariance function constant with respect to the parameters of the policy. First, we consider the case where its covariance matrix decreases during the optimization through a manual scheduling. In this context, under property 5.1, there exists an original deterministic policy and the covariance of the continuation decreases through the optimization, such that the policy-gradient algorithm optimizes this policy by continuation. Second, we consider the case where the entropy is regularized with a decreasing regularization term (e.g., by scheduling the Lagrange multiplier). Then, as entropy regularization can be seen as a constraint on the covariance of the policy, under property 5.2, there exists an original Gaussian policy and the covariance of the continuation decreases through the optimization, such that the policy-gradient algorithm optimizes this stochastic policy by continuation. Finally, as stated previously and shown in Theorem 2 in Appendix A, optimizing the return of the mirror policy of another mirror policy is equivalent to optimizing a continuation of the original policy. Therefore, policy-gradient algorithms that optimize affine Gaussian policies with both discounted covariance and decreasing regularization by local search can also be interpreted as algorithms optimizing the mean function (i.e., a deterministic policy) of this policy by continuation.

We now illustrate how policy-gradient algorithms implicitly optimize by continuation. We take as example an environment in which a car moves in a valley and must reach its lowest point (positioned in $x_{target}$) to maximize the expected sum of rewards gathered by the agent, see Appendix B. We assume we want to find the best K-controller, i.e., a deterministic policy $\mu_\theta(x) = \theta \times (x - x_{target})$, where $x$ is the position of the car. Directly optimizing such a policy is in practice subject to converging to a local extremum, as explain hereafter. We thus consider the Gaussian policy $\pi_\theta^{GP}(a|x) = \mathcal{N}(a|\mu_\theta(x), \sigma')$, where $\mu_\theta(x)$ and $\sigma'$ are the mean and variance of the policy, respectively. This policy is a mirror policy of the deterministic policy $\mu_\theta$ under a continuation of variance $\lambda = \sigma'/(x - x_{target})^2$, see Property 4.3. As can be seen in Figure 1, for each value of $\sigma'$, the return of the mirror policy equals the smoothed return of the original deterministic policy $\mu_\theta$. Consequently, optimizing by policy gradient the Gaussian policy is equivalent to optimizing the deterministic policy by continuation. For a well-chosen sequence of $\sigma'$, with a fixed scheduling or with adequate entropy regularization, the successive solutions found by local search will escape the basin of attraction of the suboptimal parameter for any initial parameter of the local search – whereas optimizing the deterministic policy directly would provide suboptimal solutions.

In this section, we have established an equivalence between the optimization of some policies by policy gradient and the optimization of an underlying policy by continuation. It opens up new questions about the hypothesis space of the (mirror) policy to consider in practice in order to exploit the properties of continuations at best. These considerations are made in the next section. We finally recall that a central assumption in the previous results is the affinity of policies. Seemingly restrictive, such policies allow to optimally control complex environments in practice (Rajeswaran et al., 2017) and give first-order results for non-affine policies.

## 5.2 Continuations for Interpreting Stochastic Policies

In practice, we know that optimizing stochastic policies tends to converge to a final policy with low variance and better performance than if we had directly optimized a deterministic policy. Practitioners often justify this observation by the need to explore through a stochastic policy. Nevertheless, to our knowledge, this concept inherited from bandit theory is not well defined for direct policy optimization. The previous

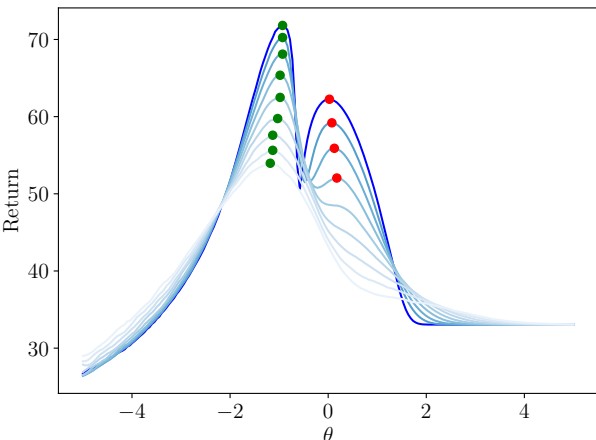

Figure 1: Illustration of the return of the policies $\mathcal{N}(a|\mu_\theta(x), \sigma')$, where $\mu_\theta(x) = \theta \times (x - x_{target})$, for different $\sigma'$ values. The darker the curve, the smaller $\sigma'$, and the darkest one is the return of the deterministic policy $\mu_\theta$. The green dots represent the global maxima and the red dots the local maxima. For some sufficiently large value for $\sigma'$, the return of the policy has a single extremum. For a well chosen schedule of decreasing $\sigma'$, a local-search algorithm will track the sequence of global extrema and converge towards the optimal deterministic policy.

analysis establishes an equivalence between optimizing stochastic policies with policy-gradient algorithms and optimizing deterministic policies by continuation. Furthermore, as explained in Section 3.2, the continuation equation (8) consists in smoothing the return of this deterministic policy through the continuation distribution. Local optima tend to be removed when the variance of the continuation is sufficiently large. Optimizing stochastic policies and regularizing the entropy, as in most state-of-the-art policy-gradient algorithms, is therefore expected to avoid local extrema before converging towards policies with small variance. We thus provide a theoretical motivation for the performance reached by algorithms applying exploration as understood in direct policy optimization.

The relationships between optimization by continuation and policy gradient in Section 5.1 have been established relying on Property 4.2 and Property 4.3. They assume continuations where the covariance matrix depends only on the current state and not on the whole observed history. In the general case, Property 4.4 allows one to extend these results by performing an analysis similar to Section 5.1. To be more specific, let us assume an affine Gaussian policy $\pi_\theta^{GP'}$, where the mean $\mu_\theta$ is a function of the state and where the covariance $\Sigma_\theta = \Sigma$ is a function of the history and is constant with respect to $\theta$. Under this assumption, if $d_{\mathcal{A}} \leq d_\Theta$ and $\nabla_\theta \mu_\theta(s)$ is full rank, the return of the policy $\pi_\theta^{GP'}$ is equal to a (unknown) continuation of the mean function $\mu_\theta$ (i.e., a deterministic policy). Furthermore, optimizing the Gaussian policy by policy gradient while discounting the covariance can be interpreted as optimizing the deterministic policy $\mu_\theta$ by continuation. In practice, this result suggests to optimize history-dependent policies by policy gradient to take advantage of the most general regularization of the objective function through implicit continuation. A similar observation was recently made by Mutti et al. (2022) who argued that history-dependent policies are required when more complex regularizations are involved. In parallel, Patil et al. (2022) also discussed the potential advantage of using history-depend parameterized policies for approximating correctly optimal policies using little representation power.

Finally, a last point has been left open in the previous discussions, namely the update of the covariance matrix of the mirror policies. The latter is defined through the covariance of the continuation. Therefore, the covariance must decrease through the optimization and must be chosen to avoid local optima. One direction to investigate in order to select a variance that removes local extrema is to update the parameters of the policy by following a combination of two directions: the functional gradient of the optimized policy's return with respect to the policy mean and the functional gradient of another measure (to be defined) with respect to the policy variance. An example of heuristic measure for smoothness might be the entropy of the

actions and/or states encountered in histories. This strategy obviously does not follow the classical approach when optimizing stochastic policies where the covariance is adapted by the policy-gradient algorithm to locally maximize the return and the exact procedure for updating the variance will require future studies. However the empirical inefficiency of this classical approach has already been highlighted in previous works that improved the performance of policy-gradient algorithms by exploring alternative learning objective functions (Houthooft et al., 2018; Papini et al., 2020).

## 6 Conclusion

In this work, we have studied the problem formulation, i.e., policy parameterization and reward-shaping strategy, when solving direct policy optimization problems. More particularly, we established connections between formulations of state-of-the-art policy-gradient algorithms and the optimization by continuation framework (Allgower & Georg, 1980). We have shown that algorithms optimizing stochastic policies and regularizing the entropy inherit the properties of optimization by continuation and are thus less subject to converging towards local optima. In addition, the role of the variance of the policies is reinterpreted in this framework: it is a parameter of the optimization procedure to adapt in order to avoid local extrema. Additionally, to inherit the properties of generic continuations, it may be beneficial to consider variances that are functions of the history of states and actions observed at each time step.

Our study leaves several questions open. Firstly, our results rely on several assumptions that may not hold in practice. Specifically, it is unclear how our findings can be generalized to non-affine policies and alternative to Gaussian policies. Nonetheless, our results can be extended in cases where we can obtain an analytic expression for the mirror policy outlined in Theorem 1. While finding such an expression may be challenging in general, we can easily extend our conclusions to non-affine policies by considering the first-order approximation. Additionally, our study is focused on Gaussian policies, which are commonly used in continuous state-action spaces. However, for discrete action spaces, a natural choice of policy is a Bernoulli distribution over the actions (or a categorical distribution for more than one action). If the state space is also discrete, this distribution may be parameterize by a table providing the success probability of the Bernoulli distribution for each state. In the case of a Beta continuation distribution, a mirror policy can be derived where actions follow a Beta-binomial distribution in each state, a result known in Bayesian inference as the Beta distribution is a conjugate distribution of the binomial distribution (Bishop & Nasrabadi, 2006). An analysis of this mirror policy would allow us to draw conclusions equivalent to those of the continuous case studied in this paper. Secondly, the study focused on entropy regularization of the policy only. Recent works have underlined the benefits of other regularization strategies that enforce the spread of other distributions as the state visiting frequency or the marginal state probability (Hazan et al., 2019; Guo et al., 2021; Mutti et al., 2022). Future research is also needed to better understand the effect of these regularizations on the optimization procedure.

Finally, we give a new interpretation for the variance of policies that suggests it shall be updated to avoid local extrema rather than to maximize the return locally. A first strategy for updating the variance is proposed in Section 5.2, which opens the door to further research and new algorithm development.

## 7 Acknowledgments

The authors would like to thank Csaba Szepesvári for the discussion on some mathematical aspects that allowed to increase the quality of this study. We also thank our colleagues Gaspard Lambrechts, Arnaud Delaunoy, Pascal Leroy, and Bardhyl Miftari for valuable comments on this manuscript. Adrien Bolland gratefully acknowledges the financial support of a research fellowship of the F.R.S.-FNRS.

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

## A    Theoretical Results on Mirror Policies

**Theorem 1.**    *For any original history-dependent policy $\eta_\theta \in \mathcal{E}$ parameterized with the vector $\theta \in \mathbb{R}^{d_\Theta}$ and for any continuation distribution $q$ and covariance function $\Lambda$, there exists a mirror history-dependent policy $\eta'_\theta \in \mathcal{E}$ of the original policy $\eta_\theta$ that writes as:*

$$\eta'_\theta(a|h) = \mathop{\mathbb{E}}_{\theta' \sim q(\cdot|\theta, \Lambda(h))} [\eta_{\theta'}(a|h)] . \tag{14}$$

**Proof.**    The continuation $f^q_\Lambda$ is defined in equation (8) as the expectation of the function (7) over the distribution (8) such that:

$$f^q_\Lambda(\theta) = \mathop{\mathbb{E}}_{\substack{s_0 \sim p_0(\cdot) \\ \theta_t \sim q(\cdot|\theta, \Lambda(h_t)) \\ a_t \sim \eta_{\theta_t}(\cdot|h_t) \\ s_{t+1} \sim p(\cdot|s_t, a_t)}} \left[ \sum_{t=0}^{\infty} \gamma^t \rho(s_t, a_t) \right] \tag{15}$$

$$= \int \left( p(s_0) \prod_{t=0}^{\infty} \eta_{\theta_t}(a_t|h_t) q(\theta_t|\theta, \Lambda(h_t)) p(s_{t+1}|s_t, a_t) \right) \left( \sum_{t=0}^{\infty} \gamma^t \rho(s_t, a_t) \right) ds_0 da_0 d\theta_0 \dots . \tag{16}$$

For the sake of simplifying the notations, let $h = (s_0, a_0, s_1, a_1, \dots) \in H$ be a history and let $R(h)$ be the discounted sum of rewards computed from this history. Let us reorder the terms of the integral and change the order of integration such that:

$$f^q_\Lambda(\theta) = \int \left( p(s_0) \prod_{t=0}^{\infty} \eta_{\theta_t}(a_t|h_t) q(\theta_t|\theta, \Lambda(h_t)) p(s_{t+1}|s_t, a_t) \right) R(h) \, dh d\theta_0 \dots \tag{17}$$

$$= \int \left( \prod_{t=0}^{\infty} \eta_{\theta_t}(a_t|h_t) q(\theta_t|\theta, \Lambda(h_t)) \right) \left( p(s_0) \prod_{t=0}^{\infty} p(s_{t+1}|s_t, a_t) \right) R(h) \, dh d\theta_0 \dots \tag{18}$$

$$= \int \left( \int \prod_{t=0}^{\infty} \eta_{\theta_t}(a_t|h_t) q(\theta_t|\theta, \Lambda(h_t)) d\theta_0 \dots \right) \left( p(s_0) \prod_{t=0}^{\infty} p(s_{t+1}|s_t, a_t) \right) R(h) \, dh . \tag{19}$$

In the inner integral over the parameters, each term of the product depends solely on the parameter at a single time step such that the integral of the product simplifies to the product of the integrals as follows:

$$f^q_\Lambda(\theta) = \int \left( \prod_{t=0}^{\infty} \int \eta_{\theta_t}(a_t|h_t) q(\theta_t|\theta, \Lambda(h_t)) \, d\theta_t \right) \left( p_0(s_0) \prod_{t=0}^{\infty} p(s_{t+1}|s_t, a_t) \right) R(h) \, dh \tag{20}$$

$$= \int \left( \prod_{t=0}^{\infty} \eta'_\theta(a_t|h_t) \right) \left( p_0(s_0) \prod_{t=0}^{\infty} p(s_{t+1}|s_t, a_t) \right) R(h) \, dh . \tag{21}$$

By definition, the latter equation is equal to the return $J(\eta'_\theta)$ of the policy $\eta'_\theta$ for any parameter vector $\theta$. Therefore, $\eta'_\theta$ is a mirror policy of the original policy $\eta_\theta$ under the process $q$ and covariance matrix $\Lambda$.

$\square$

**Theorem 2.**    *Let $q$ be a continuation distribution and let $\Lambda$ be a covariance function as defined in Section 3.2. In addition, let $\eta_\theta$, $\eta'_\theta$ and $\eta''_\theta$ be three parameterized history-dependent policies such that:*

$$\eta'_\theta(a|h) = \int \eta_{\theta'}(a|h) q(\theta'|\theta, \Lambda(h)) \, d\theta' \tag{22}$$

$$\eta''_\theta(a|h) = \int \eta'_{\theta''}(a|h) q(\theta''|\theta, \Lambda(h)) \, d\theta'' . \tag{23}$$

*Then, $\eta'_\theta$ is a mirror policy of the original policy $\eta_\theta$ and $\eta''_\theta$ is a mirror policy of the original policy $\eta'_\theta$ under continuation distribution $q$ and covariance function $\Lambda$. In addition, there exists a continuation for which $\eta''_\theta$ is a mirror policy of the original policy $\eta_\theta$.*

**Proof.** First, $\eta'_\theta$ is a mirror policy of the original policy $\eta_\theta$ and $\eta''_\theta$ is a mirror policy of the original policy $\eta'_\theta$ under continuation distribution $q$ and covariance function $\Lambda$, see Theorem 1. Then, let us substitute equation (22) in equation (23):

$$\eta''_\theta(a|h) = \int \eta'_{\theta''}(a|h) q(\theta''|\theta, \Lambda(h)) \, d\theta'' \tag{24}$$

$$= \int \left( \int \eta_{\theta'}(a|h) q(\theta'|\theta'', \Lambda(h)) \, d\theta' \right) q(\theta''|\theta, \Lambda(h)) \, d\theta'' \tag{25}$$

$$= \int \eta_{\theta'}(a|h) \left( \int q(\theta'|\theta'', \Lambda(h)) q(\theta''|\theta, \Lambda(h)) \, d\theta'' \right) d\theta' \, . \tag{26}$$

We thus have that:

$$\eta''_\theta(a|h) = \int \eta_{\theta'}(a|h) p_\theta(\theta'|h) \, d\theta' \tag{27}$$

$$p_\theta(\theta'|h) = \int q(\theta'|\theta'', \Lambda(h)) q(\theta''|\theta, \Lambda(h)) \, d\theta'' \, . \tag{28}$$

The distribution $p_\theta$ is a continuation distribution with a spread depending on the history $h$ through the covariance function $\Lambda$. By Theorem 1, $\eta''_\theta$ is a mirror policy of the original policy $\eta_\theta$.

$\square$

**Property 4.1.** *Let the original policy $\pi_\theta \in \Pi$ be a Markov policy and let the covariance function depend solely on the last state in the history. Then, there exists a mirror Markov policy $\pi'_\theta \in \Pi$.*

**Proof.** By hypotheses, the covariance matrix only depends on the last state $s_t$ of the history $h_t$, therefore:

$$q(\theta_t|\theta, \Lambda(h_t)) = q(\theta_t|\theta, \Lambda(s_t)) \, . \tag{29}$$

In addition, the original policy $\pi_\theta$ is a Markov policy, therefore :

$$\eta_\theta(a_t|h_t) = \pi_\theta(a_t|s_t) \, . \tag{30}$$

The closed form of the mirror policy, provided by equation (14), can thus be simplified as:

$$\eta'_\theta(a_t|h_t) = \int \eta_{\theta_t}(a_t|h_t) q(\theta_t|\theta, \Lambda(h_t)) \, d\theta_t \tag{31}$$

$$= \int \pi_{\theta_t}(a_t|s_t) q(\theta_t|\theta, \Lambda(s_t)) \, d\theta_t \, . \tag{32}$$

The previous equation is independent of $h_t$ knowing $s_t$, there thus exists a Markov mirror policy $\pi'_\theta \in \Pi$ respecting Theorem 1 such that:

$$\eta'_\theta(a_t|h_t) = \pi'_\theta(a_t|s_t) \, . \tag{33}$$

$\square$

**Property 4.2.** *Let the original policy $\pi_\theta^{GP} \in \Pi$ be a Gaussian policy as defined in equation (3) with affine function approximators. Let the covariance function depend solely on the last state in the history and let the distribution $q$ be a Gaussian distribution. Then, there exists a mirror Markov policy $\pi'_\theta \in \Pi$ such that for all states $s \in \mathcal{S}$, it converges towards a Gaussian policy in the limit as the affine coefficients of the covariance matrix $\Sigma_\theta(s)$ approaches zero ($\|\nabla_\theta \Sigma_\theta(s)\| \to 0$):*

$$\pi'_\theta(a|s) \to \mathcal{N}(a|\mu_\theta(s), \Sigma'_\theta(s)) \, , \tag{34}$$

*where $\Sigma'_\theta(s) = C_\theta(s) + \Sigma_\theta(s)$ and $C_\theta(s) = \nabla_\theta \mu_\theta(s)^T \, \Lambda(s) \, \nabla_\theta \mu_\theta(s)$.*

**Proof.** First, the existence of a Markov mirror policy results from Property 4.1 and is provided by equation (33):

$$\pi'_\theta(a_t|s_t) = \int \pi_{\theta_t}(a_t|s_t) q(\theta_t|\theta, \Lambda(s_t)) \, d\theta_t \; . \tag{35}$$

In addition, $\pi_{\theta_t}$ and $q$ are Gaussian distributions by hypotheses:

$$\pi_{\theta_t}(a_t|s_t) = \mathcal{N}(a_t|\mu_{\theta_t}(s_t), \Sigma_{\theta_t}(s)) \tag{36}$$

$$q(\theta_t|\theta, \Lambda(s_t)) = \mathcal{N}(\theta_t|\theta, \Lambda(s_t)) \; , \tag{37}$$

where $\mu_{\theta_t}(s_t)$ and $\Sigma_{\theta_t}(s_t)$ are affine functions of $\theta_t$. Therefore, these functions can be written as follows:

$$\mu_{\theta_t}(s_t) = \left(\nabla_{\theta_t}\mu_{\theta_t}(s_t)\right)\theta_t + \mu'(s_t) \tag{38}$$

$$\Sigma_{\theta_t}(s_t) = \left(\nabla_{\theta_t}\Sigma_{\theta_t}(s_t)\right)\theta_t + \Sigma'(s_t) \; . \tag{39}$$

For any state $s_t$, in the limit as affine coefficients of the covariance approaches zero, the covariance is such that:

$$\lim_{\|\nabla_{\theta_t}\Sigma_{\theta_t}(s_t)\|\to 0} \Sigma_{\theta_t}(s_t) = \Sigma'(s_t) \; . \tag{40}$$

In this limit, equation (35) consists in marginalizing a conditional linear Gaussian transition model with a Gaussian prior and is such that (Bishop & Nasrabadi, 2006):

$$\lim_{\|\nabla_\theta \Sigma_\theta(s_t)\|\to 0} \pi'_\theta(a_t|s_t) = \mathcal{N}\left(a_t|\left(\nabla_\theta\mu_\theta(s_t)\right)\theta + \mu'(s_t), \left(\nabla_\theta\mu_\theta(s_t)\right)^T\Lambda(s_t)\left(\nabla_\theta\mu_\theta(s_t)\right) + \Sigma'(s_t)\right) \tag{41}$$

$$= \mathcal{N}\left(a_t|\mu_\theta(s_t), \left(\nabla_\theta\mu_\theta(s_t)\right)^T\Lambda(s_t)\left(\nabla_\theta\mu_\theta(s_t)\right) + \Sigma_\theta(s_t)\right) \; . \tag{42}$$

$\square$

**Property 4.3.** *Let the original policy $\mu_\theta \in M$ be an affine deterministic policy. Let the covariance function depend solely on the last state in the history and let the distribution $q$ be a Gaussian distribution. Then, the Markov policy $\pi_\theta^{GP'} \in \Pi$ is a mirror policy:*

$$\pi_\theta^{GP'}(a|s) = \mathcal{N}(a|\mu_\theta(s), \Sigma'_\theta(s)) \; , \tag{43}$$

*where $\Sigma'_\theta(s) = \nabla_\theta\mu_\theta(s)^T \Lambda(s) \nabla_\theta\mu_\theta(s)$.*

**Proof.** The statement results from the particularization of Property 4.2 to the case of deterministic policies. Let $\pi_\theta^{GP} \in \Pi$ be an affine Gaussian policy with constant covariance matrix for any state $\Sigma_\theta(s_t) = C$. In that case, we have by Property 4.2 that $\pi'_\theta \in \Pi$ is a mirror policy as follows:

$$\pi'_\theta(a_t|s_t) = \mathcal{N}\left(a_t|\mu_\theta(s_t), \left(\nabla_\theta\mu_\theta(s_t)\right)^T\Lambda(s_t)\left(\nabla_\theta\mu_\theta(s_t)\right) + \Sigma_\theta(s_t)\right) \tag{44}$$

$$= \mathcal{N}\left(a_t|\mu_\theta(s_t), \left(\nabla_\theta\mu_\theta(s_t)\right)^T\Lambda(s_t)\left(\nabla_\theta\mu_\theta(s_t)\right) + C\right) \; . \tag{45}$$

Taking the limit of $\pi_\theta^{GP}$ as the constant covariance matrix $C$ approaches zero, we get that the original policy from Property 4.2 converges to the one of Property 4.3, namely the deterministic policy $\mu_\theta$. This implies that the policy $\pi_\theta^{GP'} \in \Pi$ provided by the limit of the mirror policy from Property 4.2, see equation (45), is a mirror policy of the original policy $\mu_\theta$ from Property 4.3:

$$\pi_\theta^{GP'}(a_t|s_t) = \lim_{C\to 0} \pi'_\theta(a_t|s_t) = \mathcal{N}\left(a_t|\mu_\theta(s_t), \left(\nabla_\theta\mu_\theta(s_t)\right)^T\Lambda(s_t)\left(\nabla_\theta\mu_\theta(s_t)\right)\right) \; . \tag{46}$$

$\square$

**Property 4.4.** *Let the original policy $\mu_\theta \in M$ be an affine deterministic policy. Let the distribution $q$ be a Gaussian distribution. Then, the policy $\eta'_\theta \in \mathcal{E}$ is a mirror policy:*

$$\eta'_\theta(a|h) = \mathcal{N}(a|\mu_\theta(s), \Sigma'_\theta(h)) , \tag{47}$$

*where $\Sigma'_\theta(h) = \nabla_\theta \mu_\theta(s)^T \Lambda(h) \nabla_\theta \mu_\theta(s)$.*

**Proof.** The policy $\mu_{\theta_t}$ is an affine function of the parameter vector $\theta_t$ and can thus be written as follows:

$$\mu_{\theta_t}(s_t) = (\nabla_{\theta_t} \mu_{\theta_t}(s_t)) \theta_t + \mu'(s_t) . \tag{48}$$

In addition, the samples drawn from the process $q$ are distributed according to a Gaussian distribution:

$$q(\theta_t|\theta, \Lambda(h_t)) = \mathcal{N}(\theta_t|\theta, \Lambda(h_t)) . \tag{49}$$

The closed form of the density of the mirror policy, provided by equation (14), is thus simplified as:

$$\eta'_\theta(a_t|h_t) = \int \eta_{\theta_t}(a_t|h_t) q(\theta_t|\theta, \Lambda(h_t)) \, d\theta_t \tag{50}$$

$$= \int \eta_{\theta_t}(a_t|h_t) \mathcal{N}(\theta_t|\theta, \Lambda(h_t)) \, d\theta_t , \tag{51}$$

where $\eta_{\theta_t}$ is the policy where each action respecting equation (48) has a probability one. The policy is a degenerated Gaussian distribution (Rao, 1973), it provides a dirac measure to each state, and its (generalized) density function may be approached as follows:

$$\eta_{\theta_t}(a_t|h_t) = \lim_{\|\Sigma\| \to 0} \mathcal{N}(a_t| (\nabla_{\theta_t} \mu_{\theta_t}(s_t)) \theta_t + \mu'(s_t), \Sigma) . \tag{52}$$

By substitution, we therefore get that the mirror policy $\eta'_\theta$ writes as follow:

$$\eta'_\theta(a_t|h_t) = \int \eta_{\theta_t}(a_t|h_t) \mathcal{N}(\theta_t|\theta, \Lambda(h_t)) \, d\theta_t \tag{53}$$

$$= \int \lim_{\|\Sigma\| \to 0} \mathcal{N}(a_t| (\nabla_{\theta_t} \mu_{\theta_t}(s_t)) \theta_t + \mu'(s_t), \Sigma) \mathcal{N}(\theta_t|\theta, \Lambda(h_t)) \, d\theta_t . \tag{54}$$

The product of the Gaussian prior over parameters and the linear Gaussian transition model of the actions provides a joint Gaussian distribution of actions and parameters (Bishop & Nasrabadi, 2006), which is degenerated but has a density for the (marginal) Gaussian distribution of actions (Rao, 1973). The density of the mirror policy $\eta'_\theta$ can thus be computed taking the limit of the marginalization:

$$\eta'_\theta(a_t|h_t) = \lim_{\|\Sigma\| \to 0} \int \mathcal{N}(a_t| (\nabla_{\theta_t} \mu_{\theta_t}(s_t)) \theta_t + \mu'(s_t), \Sigma) \mathcal{N}(\theta_t|\theta, \Lambda(h_t)) \, d\theta_t \tag{55}$$

$$= \lim_{\|\Sigma\| \to 0} \mathcal{N}\left(a_t| (\nabla_\theta \mu_\theta(s_t)) \theta + \mu'(s_t), (\nabla_\theta \mu_\theta(s_t))^T \Lambda(h_t) (\nabla_\theta \mu_\theta(s_t)) + \Sigma\right) \tag{56}$$

$$= \lim_{\|\Sigma\| \to 0} \mathcal{N}\left(a_t|\mu_\theta(s_t), (\nabla_\theta \mu_\theta(s_t))^T \Lambda(h_t) (\nabla_\theta \mu_\theta(s_t)) + \Sigma\right) \tag{57}$$

$$= \mathcal{N}\left(a_t|\mu_\theta(s_t), (\nabla_\theta \mu_\theta(s_t))^T \Lambda(h_t) (\nabla_\theta \mu_\theta(s_t))\right) . \tag{58}$$

We note that this result can be obtained without working on degenerated Gaussian distributions. The policy is an affine function of the parameters, which follow a Gaussian distribution, the marginal distribution of actions is thus also a Gaussian distribution of the form of equation (58). This distribution is furthermore the one of a mirror policy, see Theorem 1.

$\square$

# B  Description of the Car Environment

In this section, we formalize the reinforcement learning environment that models the movement of a car in a valley with two floors separated by a peak, as depicted in Figure 2. The car always starts at the topmost floor and receives rewards proportionally to its depth in the valley. An optimal agent drives the car from the initial position to the lowest floor in the valley by passing the peak. In the following, we describe each element composing the environment.

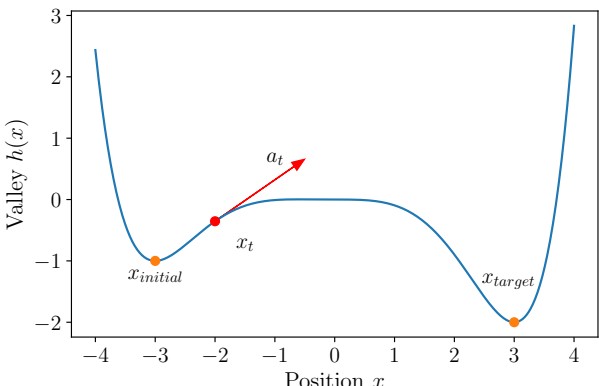

Figure 2: Valley in which the car moves.

**State Space.**  The state $s_t \in \mathbb{R}^2$ of the environment is composed of two scalar values, namely the position $x_t \in \mathbb{R}$ of the pointmass representing the car and its tangent speed $v_t \in \mathbb{R}$.

**Action Space.**  At each time step, the agent controls the force applied on the car through the actions $a_t \in \mathbb{R}$ it executes.

**Initial Position.**  The car always starts at the topmost floor $x_{initial} = -3$ in the valley at rest. The initial state distribution thus provides a probability one to the state

$$x_0 = x_{initial} \tag{59}$$
$$v_0 = 0 \,. \tag{60}$$

**Transition Distribution.**  The continuous motion of the car in the valley is derived for Newton's formula. The valley's analytical description is provided by the function $h$, the car's mass is denoted by $m = 0.5$, gravitational acceleration by $g = 9.81$, and damping factor by $e = 0.65$. The position $x$ and speed $v$ of the car follow the subsequent continuous-time dynamics as a function of the force $a$:

$$\dot{x} = v \tag{61}$$
$$\dot{v} = \frac{a}{m(1 + h'(x)^2)} - \frac{gh'(x)}{1 + h'(x)^2} + \frac{v^2 h'(x) h''(x)}{1 + h'(x)^2} - ev^2 \,. \tag{62}$$

The position and force are furthermore bounded to intervals as part of the dynamics such that

$$x \in [x_m, x_M] = [-4, 5] \tag{63}$$
$$a \in [a_m, a_M] = [-10, 10] \,. \tag{64}$$

Clamped force values are therefore used in equation (62). Similarly, the position is clamped in equation (61).

In discrete time, the state $s_{t+1}$ is computed through Euler integration of the continuous-time dynamics, considering an initial position given by the current state $s_t$. The force $a$ remains constant during a discretization time $\Delta = 0.1$ and is equal to the action $a_t$, with an additive noise drawn from $\mathcal{N}(\cdot|0, 1)$ and clamped before integration.

**Reward Function.** The rewards correspond to the depth of the valley at the current position. The reward function thus solely depends on the position

$$\rho(s_t, a_t) = -h(x_t) \ . \tag{65}$$

**Discount Factor.** The discount factor equals $\gamma = 0.99$ and the horizon is curtailed to $T = 100$ in each numerical computation.

