# OpenReview forum: "Policy Gradient Algorithms Implicitly Optimize by Continuation"
_TMLR — Accepted by TMLR_

### Review · Reviewer_EhvW · 2023-06-10

**Summary Of Contributions:**

The submission provides an interpretation of optimizing a Gaussian policy instead of a deterministic policy or adding entropy regularization as optimization by continuation (aka graduated optimization), a global search method that smoothens the objective function at monotonically decreasing levels to avoid bad local optima. An important difference between the analysis in this paper and common policy gradient algorithms is that here the covariance matrix of the Gaussian policy is not optimized but should be annealed independently to match the framework of optimization by continuation.

**Audience:**

Yes

**Claims And Evidence:**

Yes

**Requested Changes:**

The comments in the Correctness section are all critical.

**Strengths And Weaknesses:**

The final decision is left for after discussions. The scope of the paper is appropriate for TMLR. The connection to optimization by continuation is intuitive and is explained in an example environment. The proofs for two of the main results are not presented clearly (comments 1 and 2 in Correctness Section below) and I can only assess the correctness of the results if the authors clarify these proofs in a revision.

Correctness:
1. There are multiple steps to obtain Eq 23 from Eq 22 in the appendix which should be laid out. In particular I do not understand how the order of the product over t and the expectation over theta are swapped to obtain Eq 23.

2. The discussion for Gaussian policies in Section 5.1 should be stated formally, ideally as propositions. Why does the condition on dimensions and the rank of the gradient satisfy the requirement in this discussion? What does it precisely mean that the covariance is discounted and why does it guarantee full rank gradient?

3. Section 2.2 mentions that an MDP always has an optimal deterministic policy. This is true for *finite* MDPs. A continuous-action MDP, the problem this paper focuses on, may not have an optimal policy in general (stochastic or deterministic). I recommend removing this statement.

4. The same paragraph cites Silver et al (2014) to claim deterministic policies tend to get stuck in local optima. Silver et al in fact mention inadequate exploration as the problem with deterministic policies.

5. The conclusion mentions that a categorical distribution is a natural policy parametrization for discrete state space. The action space, and not the state space, determines policy parametrization. A continuous-action discrete-state problem would still naturally use Gaussian or deterministic (rather than categorical) policies.

6. Does the inner loop in the optimization by continuation algorithm continue until convergence or for one or a few steps? If it is until convergence, this difference between this algorithm and policy gradient algorithms should be clarified in the text.

Organization:
1. The results that follow from Theorem 1 are presented as "Property 1", "Property 2", etc. except Theorem 2 which is called a theorem, described informally in the main paper, and proven in a dedicated section in the appendix before the proof for Theorem 1. I think presenting the result like the others would make the paper clearer. The paper is otherwise well organized.

Notation and typos:
1. Even if "p" is the conventional notation for continuation it's still better to use a different letter here since p is already used for start distribution and transition kernel.
2. Extra phrase "algorithm algorithmic" before Algorithm 1.
3. In all references to equations "equation" is repeated.
4. (Below property 1) Independently *of* the history.
5. From property 2 onwards the notation for covariance matrix changes to have a subscript of theta.
6. (Section 5.1, Gaussian policies) "does optimizing a Gaussian policy is" -> "optimizing a Gaussian policy is". "trough" -> "through"

---

> ### Author Response · Authors · 2023-06-15
> **Response to Reviewer EhvW - part 1**
>
> Dear reviewer, thank you for your valuable comments. We have revised the manuscript accordingly, and highlighted the main changes in yellow for ease of review. Please find our point-by-point responses below. We hope that these changes address your concerns and improve the overall quality of the manuscript.
>
> Best regards,
>
> The authors
>
> > 1 . There are multiple steps to obtain Eq 23 from Eq 22 in the appendix which should be laid out. In particular I do not understand how the order of the product over t and the expectation over theta are swapped to obtain Eq 23.
>
> We have included additional intermediate steps in the proof provided in the appendix. In particular, we have explained how it is possible to transform the integral over a product into a product of integrals in the transition from equation (19) to (20), in the revised version. This transformation is valid because the variables of integration appear independently in each term of the product.
>
> > 2 . The discussion for Gaussian policies in Section 5.1 should be stated formally, ideally as propositions. Why does the condition on dimensions and the rank of the gradient satisfy the requirement in this discussion? What does it precisely mean that the covariance is discounted and why does it guarantee full rank gradient?
>
> The condition on dimensions and rank are necessary for ensuring the existence of original policies and continuations such that the policies optimized by policy gradients in both formulations are mirror policies of these original policies under the specified continuations. Moreover, the equivalence between the optimization by policy-gradient algorithms and by continuation require an additional assumption. For the first formulation, the variance of the mirror policies, the one optimized by policy gradient, must decrease during optimization. This can be guaranteed by manually scheduling the variance to be decreasing. The three assumptions are complementary, and none implies the others in a general case. Section 5.1 has been reworked to make the elements of the discussion explicit in the form of properties. The discussion of the different assumptions for the equivalence is also better segmented, in different paragraphs.
>
> > 3 . Section 2.2 mentions that an MDP always has an optimal deterministic policy. This is true for finite MDPs. A continuous-action MDP, the problem this paper focuses on, may not have an optimal policy in general (stochastic or deterministic). I recommend removing this statement.
>
> > 4 . The same paragraph cites Silver et al (2014) to claim deterministic policies tend to get stuck in local optima. Silver et al in fact mention inadequate exploration as the problem with deterministic policies.
>
> Thank you for these clarifications. The paragraph in question has been reworked to include your comments 3 and 4.
>
> > 5 . The conclusion mentions that a categorical distribution is a natural policy parametrization for discrete state space. The action space, and not the state space, determines policy parametrization. A continuous-action discrete-state problem would still naturally use Gaussian or deterministic (rather than categorical) policies.
>
> Indeed, the action space defines the choice of distribution of the policy over the actions. However, the complete parameterization of the policy also depends on the state space. In our case, if the action space is discrete, we use a categorical distribution for the policy, and if the state space is also discrete, we reprensent the success probabilities in a tabular form. In this way, we can apply the results from Bayesian inference as explained in the paragraph. The details were implicit, and we have reworked the paragraph in order to emphasize the necessity of the discrete state space and make all details explicit.
>
> > 6 . Does the inner loop in the optimization by continuation algorithm continue until convergence or for one or a few steps? If it is until convergence, this difference between this algorithm and policy gradient algorithms should be clarified in the text.
>
> As you have pointed out, in practice, continuations are not optimized until convergence, hence the equivalence with policy-gradient algorithms. We have clarified this point by adding a paragraph at the end of section 3.1.

---

> ### Author Response · Authors · 2023-06-15
> **Response to Reviewer EhvW - part 2**
>
> > 7 . The results that follow from Theorem 1 are presented as "Property 1", "Property 2", etc. except Theorem 2 which is called a theorem, described informally in the main paper, and proven in a dedicated section in the appendix before the proof for Theorem 1. I think presenting the result like the others would make the paper clearer. The paper is otherwise well organized.
>
> Thank you for your comment. We have restructured the appendices to present the theorems in the order in which they appear in the main text. We have kept Theorem 2 in the appendix because it is an additional element of discussion to the elements presented in section 4. Since the section is already quite heavy mathematically, we think that keeping the formal result in appendix allows us to keep the central elements in first position in the main text, and thus maintain coherence and readability. We appreciate the reviewer's acknowledgment of the paper's overall organization and hope that this modification further improves its clarity.
>
> > 8 . Even if "p" is the conventional notation for continuation it's still better to use a different letter here since p is already used for start distribution and transition kernel.
>
> We appreciate your feedback, and we agree on the substance of your comment. While we acknowledge that p is used for the continuation distribution, we would like to highlight that the transition distribution/kernel can be viewed as a (conditional) marginalization of this distribution. We believe that the context of the paper provides sufficient clarity to distinguish between the different cases. To avoid introducing additional notations, we have decided to retain the current notation. Thank you for bringing this concern to our attention.
>
> > 9 . Extra phrase "algorithm algorithmic" before Algorithm 1.
>
> > 10 . In all references to equations "equation" is repeated.
>
> > 11 . (Below property 1) Independently _of_ the history.
>
> > 12 . (Section 5.1, Gaussian policies) "does optimizing a Gaussian policy is" -> "optimizing a Gaussian policy is". "trough" -> "through"
>
> Thank you for bringing this to our attention, and we apologize for the oversight in our previous version. We appreciate your feedback, and we have now incorporated the recommended changes into the paper.
>
> > 13 . From property 2 onwards the notation for covariance matrix changes to have a subscript of theta.
>
> In the previous version, we used sigma as notation for the covariance matrix of both the continuation and the policy, with the subscript theta indicating that it referred to the policy covariance. Initially, we believed that the context would provide sufficient clarity to differentiate between the two cases. However, upon further consideration, we recognize that this notation leads to confusion. As a result, we have made revisions to enhance understanding. We now employ lambda to represent the covariance of the continuation and sigma for the policy covariance. Furthermore, we remove the subscript when the covariance is constant.

---

### Review · Reviewer_gGC6 · 2023-08-01

**Summary Of Contributions:**

This works shows a link between direct policy optimization methods (in reinforcement learning) and optimization by continuation (in non-convex optimization).

Optimization by continuation solves a non-convex optimization problem by solving a sequence of "relaxed" optimization problems, on fgunctions called continuation of the objective functions. This works shows that the continuation of the return of the policy is the return of another policy (the mirrror policy), and then that solving the return by continuation is the same as solving a policy optimization problem for the mirror policy.

Then, the authors link more specific policy gradient methods to optimization by continuation. In particular, they show that in the case of a affine-parametrized Guassian policy with decreasing variance, policy gradient is equivalent to optimization by continuation on the deterministic policy (that just outputs the mean of the Gaussian).

**Audience:**

Yes

**Broader Impact Concerns:**

No broader impact concerns (theoretical work).

**Claims And Evidence:**

Yes

**Requested Changes:**

Poposed adjustments are the answers to the Weakness section. I would suggest
 - extending the background on OC,
-   discussing the theoretical and practical properties of this method, and how they can influence RL algorithm analysis adn design.

**Strengths And Weaknesses:**

# Strengths

## Clarity

The paper is clear and well presented.

## Insight

Links between RL methods and optimization are not new, but to my knowledge this specific link was not shown before in the literature. In general, I think that linking works form two different communities is often a valuable contribution, as it can provide new understanding on these methods.

In particular, this works proposes directions to understand better entropy regularization in RL. Entropy is many modern RL algorithms, but it is not always clear what its exact function is. This is a studied question in the RL literature, and while this work does not give a definitive answer to it, I think it can be interesting to many researchers in that area. For example, it fits nicely in the direction open by [1], arguing that entropy is useful because it smooth the optimization landscape of the policy.


# Weakness

I think the work is lacking a bit of context on why these results are interested. In particular, two questions are not addressed (that are not independent):

## How and why optimization by continuation is used in classic optimization ?

When introducing OC, it would be valuable to understand why this method would be used in non-convex optimization, for example if it has a inherent theoretical or practical advantage, and maybe give one example where it would be effective. This would make the contribution clearer, because right now it is hard to see the point of this method by itself.

## What are the known convergence results on optimization by continuation, and how would they translate to RL ?

I am not a specialist of optimization, but I think it would be interesting to state what are the known theoretical properties of OC. In particular, this could be discussed in relation to section 5.1. If some policy gradient algorithms can be analyzed through OC, it would be valuable to know if some analysis tools form optimization can be applied to this analysis.

---

> ### Author Response · Authors · 2023-08-09
> **Response to Reviewer gGC6**
>
> We appreciate your feedback and have made necessary revisions to the manuscript. Changes have been highlighted in yellow for easy identification.
>
> The introduction has been expanded to provide more context about the optimization by continuation framework. In particular, as explained in this extended section, optimization by continuation offers two key benefits: it smoothens objective functions to enable gradient-based optimization techniques and enables global optimization for non-convex functions. The revised introduction includes examples of successful machine learning applications of the framework.
>
> It is important to note that theoretical guarantees for optimization by continuation methods are currently lacking in the literature, and that the design of continuations is mostly heuristic and problem specific. Furthermore, we introduce a new family of continuations for policy optimization. To the authors’ knowledge, no trivial theoretical guarantees hold. A notable point discussed in Section 3 is the connection between Gaussian smoothing and the removal of local optima, which provides insight into the effect of continuations and may have the potential to inspire further algorithmic development. Arguably the most interesting result discussed in the paper is about the selection of the sequence of continuations, defined through the update of the variance of the policy during the optimization. This update shall pursue as objective the smoothing the return instead of maximizing it locally, as usually done in the literature.

---

### Review · Reviewer_bwWD · 2023-08-05

**Summary Of Contributions:**

The submission offers a novel theoretical perspective for policy-gradient algorithms (PGA). Initially, it casts PGA within the 'optimization by continuation' framework. This approach is tailored for optimizing nonconvex functions by progressively optimizing a series of surrogate objectives known as continuations. Subsequently, it demonstrates that the process of optimizing affine Gaussian policies combined with entropy regularization can be seen as an indirect optimization of deterministic policies through continuation. Drawing from these theoretical findings, it advances that exploration in PGA amounts to determining a continuation of the current policy's return. Finally, it argues that the policy variance should be based on history rather than being put in trade-off with the policy return maximization.

**Audience:**

Yes

**Broader Impact Concerns:**

I do not have concern on any potential negative impact.

**Claims And Evidence:**

Yes

**Requested Changes:**

I find the paper good enough as it is, but I am a bit frustrated by the lack of empirical studies to support it. So my only question would be, why not have some experiments that could both serve as didactic content and as validation of the theory?

**Strengths And Weaknesses:**

Strength:
- Interesting angle to tackle regularization in policy gradient optimization.
- The work is well presented: good motivation, clear introduction, sufficient background, sound mathematical exposition.

Weaknesses:
- The work is limited to a specific model of the policies function class: the affine Gaussian policies.
- I found that the submission was lacking empirical illustration of the theoretical findings: in practice what are the similarities with the main algorithms? In which kind of MDPs do we observe differences in policy optimization?

As I said before, the submission is remarkably typo-proof, and my only minor remark will be on the choice of \eta to design policies while stochastic policies are denoted by \pi, and deterministic policies by \mu. I found that it unnecessarily overloaded the notations.

---

> ### Author Response · Authors · 2023-08-09
> **Response to Reviewer bwWD**
>
> We appreciate the reviewer’s positive feedback and remarks regarding our work.
>
> We want to emphasize that our conclusion section provides insights for expanding our discussion to a diverse range of policy classes. Concerning empirical illustration, the experiment presented in the paper aims to provide practical insights into the theory, effectively supporting our findings. We understand the reviewer’s frustration, yet creating additional experiments is complex and we believe it likely won’t yield significantly greater insight.

---

### Decision · Action_Editors · 2023-09-20

**Recommendation:** Accept as is

**Comment:**

The reviewers are happy with the improvements to the work. They unanimously voted to accept.

I know that this is an Accept as is, but I would request one very minor thing. It would be great if you could consider breaking up the very large paragraphs that were added. I can see obvious subparagraphs within those large paragraphs, and they are currently hard to read.

**Audience:**

Yes, understanding policy gradient methods is of interest to a broad segment of the community.

**Claims And Evidence:**

The paper has clear theory and a well done small experiment highlighting the phenomenon.